# Designing Vitamin D₃ Formulations: An In Vitro Investigation Using a Novel Micellar Delivery System

Min Du [1], Chuck Chang [1], Xin Zhang [2], Yiming Zhang [1], Melissa J. Radford [3], Roland J. Gahler [4], Yun Chai Kuo [1], Simon Wood [5,6,7] and Julia Solnier [1,*]

1  ISURA, Burnaby, BC V3N4S9, Canada; mdu@isura.ca (M.D.); cchang@isura.ca (C.C.); yzhang@isura.ca (Y.Z.); rkuo@isura.ca (Y.C.K.)
2  4D LABS, Simon Fraser University, 8888 University Drive, Burnaby, BC V5A 1S6, Canada; xinz@sfu.ca
3  Department of Chemistry and 4D LABS, Simon Fraser University, 8888 University Drive, Burnaby, BC V5A 1S6, Canada; melissa_radford@sfu.ca
4  Factors Group R & D, Burnaby, BC V3N4S9, Canada; rgahler@factorsgroup.com
5  School of Public Health, Faculty of Health Sciences, Curtin University, Perth, WA 6845, Australia; simonwood@shaw.ca
6  InovoBiologic Inc., Calgary, AB Y2N4Y7, Canada
7  Food, Nutrition and Health Program, University of British Columbia, Vancouver, BC V6T1Z4, Canada
*  Correspondence: jsolnier@isura.ca

**Abstract:** Vitamin D is an essential nutrient with important immunomodulatory properties. As a fat-soluble compound, Vitamin D (and its D₃ form) is immiscible with water, which presents challenges to absorption. In an in vitro setting, the current study characterizes novel micellar formulations of Vitamin D₃ designed to improve absorption. Techniques used to evaluate and compare the micellar formulations against a non-micellar formula include the following: cryo-SEM to determine morphology; laser diffraction to determine particle size and distribution; zeta potential to determine stability of the particles; solubility assays to determine solubility in water and gastrointestinal media; and Caco-2 cell monolayers to determine intestinal permeability. Results show advantageous features (particle size range in the low micrometres with an average zeta potential of $-51.56 \pm 2.76$ mV), as well as significant improvements in intestinal permeability, in one optimized micellar formula (LipoMicel®). When introduced to Caco-2 cells, LipoMicel's permeability was significantly better than the control ($p < 0.01$; ANOVA). Findings of this study suggest that the novel micellar form of Vitamin D₃ (LipoMicel) has the potential to promote absorption of Vitamin D₃. Thus, it can serve as a promising candidate for follow-up in vivo studies in humans.

**Keywords:** Vitamin D; cholecalciferol; bioavailability; Caco-2 cell-permeability; delivery systems; electron microscopy; cryo-SEM; laser diffraction; zeta potential

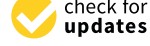



## 1. Introduction

The COVID-19 pandemic has brought increased attention to Vitamin D as an essential nutrient [1–5]. Functioning as a steroidal hormone, it is synthesized by skin cells upon sunlight irradiation, and it may also be obtained from dietary sources [6–8]. However, Vitamin D deficiency (VDD) has been a serious health issue even pre-pandemic [9]. Disorders such as autoimmune diseases (including type I diabetes mellitus and multiple sclerosis), inflammatory bowel disease, rickets, osteomalacia, and many others could result from VDD [7,8,10–12]. Unfortunately, VDD can be caused by a variety of factors that are difficult to control including the seasons, geographic locations with limited access to sunshine, ethnic characteristics (such as skin tones), dietary limitations or preferences, indoor lifestyles, and even excessive clothing [8,10,11,13]. Additionally, if a pregnant woman develops VDD, her unborn child may also be affected [8,10,11]. The immunomodulatory effects of this vitamin are unquestionably valuable in the context of epidemiology, with many reports of positive associations in the fight against COVID-19 [2,4,5].

Other than Vitamin D fortified foods, dietary sources are limited to fatty fish such as salmon, mackerel and sardines and mushrooms such as shiitake, maitake and morel. Older adults, those with dark skin, and people with certain medical conditions such as Crohn's disease, celiac disease and liver or kidney disease could be more susceptible to VDD. As a result, oral Vitamin D supplements have become highly popular and constitute a promising strategy to fight VDD [14–16] because they are readily available, cost effective and easy to administer. Most often (and possibly most beneficial), Vitamin D is supplied in the form of Vitamin $D_3$, otherwise known as cholecalciferol [17]. This form of the vitamin is used in the present study.

Ideally, orally supplemented Vitamin $D_3$ should be completely absorbed through the intestinal mucosa [18]. However, immiscibility between this hydrophobic compound and the aqueous gut environment, as well as the existence of interference from other compounds such as cholesterol, severely hinder the absorption process [11,12,18]. To overcome these challenges, creation of more bio-accessible delivery vehicles can enhance absorption and bring greater benefits of this vitamin [19].

Such vehicles for improving the absorption and efficacy of drugs have existed for decades but the concepts have only been applied to nutrients more recently [20–22]. Still, there continues to be efforts aimed at refining this technology with different materials, varying particle sizes or even imparting the vehicles with "smart" capabilities for targeted delivery [23,24]. Typically, the delivery vehicles have minute particle sizes and are expected to result in a finer dispersion of the target compound in solution, thus providing greater surface area to the enterocytes for enhanced absorption. Furthermore, delivery vehicles could impart beneficial characteristics such as added polarity or surface charges, which should theoretically assist with aqueous solubility and promote permeability through the gut-blood barrier. Liposomes are one of the most commonly used delivery systems to achieve this purpose [24]. While liposomal formulations hold promise for improved solubility and absorption [19,25], micellar formulations have also been explored, especially since their hydrophobic centres provide a more optimal condition for hydrophobic compounds such as Vitamin $D_3$ [26]. Efforts in this regard have generally been directed toward increasing solubilities by reducing the particle size of the delivery vehicles and creating nanoparticles and nanostructured lipid carriers [27–29]. However, in addition to particle size, there is increasing evidence that cell–matrix adhesion and specific receptor carriers may also play key roles in influencing absorption by using micellar delivery systems [30–32].

Thus, we aim to evaluate three novel vitamin $D_3$ formulations based on the LipoMicel® technology and hypothesized that at least one would form micellar vehicles that significantly increase intestinal absorption. LipoMicel is used to improve the solubility and stability of various compounds by encapsulating the active compound in a lipid-based matrix and creating a natural emulsion. Although other micellar formulations have been previously reported, this study focuses on the investigation of formulations using only "food grade", safe ingredients without milk allergens or the use of synthetic detergents and organic solvents [30,33–35]. It is worthwhile to investigate novel food grade formulations since they use ingredients that are safe for human consumption and are most likely to enter the market as health supplements. The in vitro analyses of these formulations serve as a preliminary screen to identify the formula that is best suited for subsequent human trials. We assessed the solubility and permeability of the novel formulations against a control, as well as morphological and physicochemical properties of the most promising formula. Emphasis is placed on data from Caco-2 permeability experiments so that the absorption contributions of the most promising formula in this biological model can be further characterized in terms of morphology and stability through cyro-SEM and zeta potential measurements. To our knowledge, studies about Vitamin D permeability using Caco-2 cells are currently scarce and this work adds to the current data of Vitamin D evidence available from this human cell line by using a novel food-grade LipoMicel formulation. Results show that one LipoMicel formulation, despite having a similar particle-size range and solubilities as the control, showed significantly improved permeability, suggesting

that more complex biological mechanisms may be involved. Overall, LipoMicel® has the potential to be effective at improving the oral absorption of Vitamin $D_3$ and stands to benefit from further in vivo experiments to determine bioavailability in humans.

## 2. Materials and Methods

### 2.1. Vitamin $D_3$ (Cholecalciferol) Formulations

Table 1 provides details on the different vitamin $D_3$ formulations examined. Vitamin $D_3$ (1,000,000 IU/g in flax oil), medium chain triglycerides (20–50% capric acid), xylitol (98.5–101%), methylsulfonylmethane (98–102%), glycerin (99–101%), stevia (>90% total steviol glycosides), lecithin (unbleached, non-GMO), cocoa (*Theobroma cacao* powder) and flaxseed oil (50–65% alpha linolenic acid) were provided by InovoBiologic (InovoBiologic Inc., Calgary, AB, Canada). Saponin (from quillaja bark, $\geq$10% sapogenin content) was obtained from Millipore-Sigma (Millipore-Sigma, Toronto, ON, Canada), and ethanol (anhydrous) is from Commercial Alcohols (Commercial Alcohols Inc., Toronto, ON, Canada). All materials were food grade. In general, 20 mL of each formulation was prepared by mixing the powdered ingredients with lipophilic components (for example, carrier oil) in a 50 mL centrifuge tube. A VWR Analogue vortex mixer (VWR International, Toronto, ON, Canada) at max setting (3200 rpm) was used to facilitate the mixing at room temperature for 5 min. Different structures or complexes such as delivery vehicles could be generated through interactions between the ingredients.

Consequently, the formula's solubility, permeability or other properties could be altered by such structures or complexes. The simplest formula, where only Vitamin $D_3$ and carrier oil (i.e., flaxseed oil) were present, was used as the baseline control (BC). Three different novel formulations (LM1, LM2 and LM3, where LM was abbreviated from "LipoMicel") were compared to BC to evaluate any changes in solubility and permeability. All formulations were provided by the Factors Group of Nutritional Companies. Common excipients were shared between LM1, LM2 and LM3 as follows: medium chain triglycerides, Xylitol, Methylsulfonylmethane. Excipients unique to each formula were the followin: glycerin, saponin and ethanol were present in LM1; stevia and lecithin were present in LM2; cocoa was present in LM3. The combinations of these are proprietary.

**Table 1.** The composition of each of the four formulations used in this study.

| Formula LM1 | Formula LM2 | Formula LM3 | Formula BC |
|---|---|---|---|
| Vitamin $D_3$ | Vitamin $D_3$ | Vitamin $D_3$ | Vitamin $D_3$ |
| Medium chain triglycerides | Medium chain triglycerides | Medium chain triglycerides | Flaxseed oil |
| Xylitol | Xylitol | Xylitol | |
| Methylsulfonylmethane | Methylsulfonylmethane | Methylsulfonylmethane | |
| Glycerin | Stevia | Cocoa | |
| Saponin | | | |
| Ethanol | Lecithin | | |

### 2.2. Solubility Analysis

Solubilities of the formulations in water, simulated gastric solution and simulated intestinal solution were analyzed. The gastrointestinal medias were prepared according to the method published by the USP.

To investigate solubility, a fixed amount of each Vitamin $D_3$ formula (containing 1500 IU or 37.5 $\mu$g of Vitamin $D_3$) was added to 1 mL of solution in a 1.5 mL centrifuge tube and allowed reach saturation. For solubilities in water, samples were vortexed to suspend visible particles and then sonicated at 37 °C for 15 min before they are transferred into 1.5 mL glass vials. For solubilities in simulated gastric and intestinal solutions, samples were vortexed and sonicated at 37 °C for 1 h before transferring to 1.5 mL glass vials. The samples were injected into an Ultra High Performance Liquid Chromatography (UHPLC) instrument to quantify dissolved cholecalciferol. Normally, samples for UHPLC analysis

are filtered before analysis, but this step was skipped to guarantee that micelles larger than the filter membrane orifices were not excluded from analysis.

### 2.3. Permeability Analysis

To assess the permeability of the formulations, Caco-2 cells (Cedarlane Laboratories, Toronto, ON, Canada) were cultured in a T-25 flask (Thermo Fisher Scientific Inc., Waltham, MA, USA) in a HERACELL VIOS 160i $CO_2$ incubator (Thermo Fisher Scientific Inc., MA, USA) set to 37 °C and 5.0% $CO_2$. The Caco-2 cell line was used as a model for the intestinal epithelial barrier, which allowed us to understand how the formulations could behave during absorption [36]. The composition of the cell culture media was as follows: Dulbecco's modified Eagle's medium (DMEM) (Sigma-Aldrich, St. Louis, MO, USA), 10% heat-inactivated fetal bovine serum (FBS) (Thermo Fisher Scientific Inc., MA, USA), penicillin (100 units/mL), and streptomycin (100 units/mL) (Sigma-Aldrich, MO, USA). Cells were resuspended with 5% trypsin (Thermo Fisher Scientific Inc., MA, USA) and seeded on a 24-well format polycarbonate semipermeable membrane insert with a diameter of 6.5 mm and a pore size of 0.4 μm (VWR International, Toronto, ON, Canada). The seeding density was $1 \times 10^4$ cells/cm². The cells were maintained for a total of 21 days before being processed for permeability assay. The media was refreshed every 48 h during the first 14 days, and every 24 h during the final seven days. An EVOM2 instrument (World Precision Instruments, Sarasota, FL, USA) was used to measure the transepithelial electrical resistance (TEER) values of the cells. Only Caco-2 monolayers with TEER values between 250–500 Ωcm² were selected for permeability experiments.

On the day of measurement, Caco-2 cells were washed twice with Hanks' balanced salt solution (HBSS) (Sigma-Aldrich, MO, USA) and then allowed to equilibrate for 30 min in the incubator at 37 °C. Thereafter, 100 μL of a Vitamin $D_3$ formula was added as the donor solution to the apical side of the monolayer, and 500 μL of HBSS was added to the basal side. Four hours after the treatment, the apical and basal solutions were collected for analysis using UHPLC. A schematic representation of the permeability measurement workflow is depicted in Figure 1. All treatments were performed in triplicates.

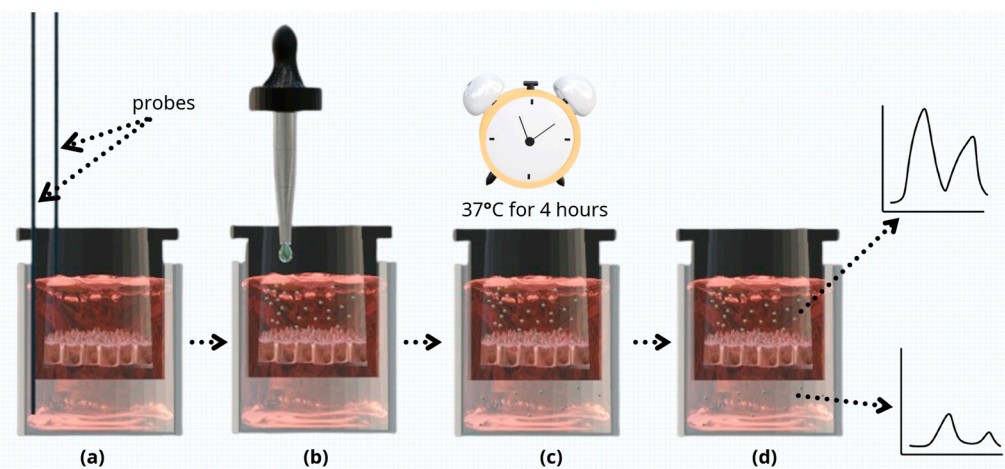

**Figure 1.** Schematic presentation of the permeability measurement workflow. (**a**) TEER values of Caco-2 monolayers were measured using EVOM2 instrument probes. Only monolayers with TEER values of 250–500 Ωcm² were selected for further analysis. (**b**) Donor solution (one of the four Vitamin $D_3$ formulations) was added to the apical compartment, and HBSS to the basal side. (**c**) Cells were incubated at 37 °C for 4 h. (**d**) Apical and basal solutions were collected and analyzed using UHPLC.

The apparent permeability coefficient ($P_{app}$, unit in cm/s) could be calculated from the permeation rate and compound concentration at $t = 4$ h (see formula below). Compound concentration was measured using Ultra High Performance Liquid Chromatography (UHPLC). In this formula, the amount of product present in the basal compartment as a function of time (nmol/s) is represented by "$dQ/dt$", the transwell area (cm²) is represented by

"*A*" and the initial concentration of the product applied in the apical compartment (μM) is represented by "$C_0$".

$$P_{app} = \frac{dQ}{dt} \cdot \frac{1}{A \cdot C_0}$$

### 2.4. Ultra High Performance Liquid Chromatography (UHPLC)

An UltiMate 3000RS UHPLC system was used. It included a quaternary pump (Thermo Fisher Scientific Inc., MA, USA), which delivered a binary gradient of 0.2% HPLC-grade phosphoric acid (VWR International, ON, Canada) in HPLC-grade water (Thermo Fisher Scientific Inc., MA, USA) and HPLC-grade methanol (Thermo Fisher Scientific Inc., MA, USA), through a Poroshell EC-18 100 × 2.1 mm, 2.7 μm column (Agilent Technologies, Santa Clara, CA, USA) at a flowrate of 0.500 mL/min. The gradient was maintained at 90% methanol over a period of 8 min. The column was equilibrated with the starting conditions for 2 min before the next injection. The column oven temperature was set to 40 °C and data were collected at a wavelength of 265 nm using a diode-array detector.

### 2.5. Particle-Size Distribution by Laser Diffraction

The formulations were mixed in water and then their particle-size distributions were determined by laser diffraction using a Mastersizer 3000 particle size analyzer (Malvern Panalytical, Quebec, QC, Canada). Briefly, samples (approximately 1 mL) were directly diluted in an Hydro SM (Malvern Panalytical, QC, Canada) wet dispersion accessory filled with approximately 200 mL of water. Once the dispersed mixture reaches 10% obscuration, data were collected over a period of 1 min while the samples are circulated through the optical measurement cell. Hydrodynamic volumes of the micelles were determined through diffraction data analyzed by the Mastersizer software 3000.

To quantitatively describe particle size distribution, the physicochemical property PDI (polydispersity index) was used. PDI was calculated using the following formula:

$$\text{PDI} = (\sigma/\mu)^2$$

where "$\sigma$" represents the standard deviation and "$\mu$" represents the mean particle diameter [37]. In the context of lipid carrier delivery systems, a PDI less than 0.3 can describe particles homogeneous in size (monodisperse), whereas higher PDI values would indicate the presence of a more varied size range (polydisperse) [34,38,39].

### 2.6. Cryo-SEM (Cryogenic Scanning Electron Microscopy)

To prepare samples for cryo-SEM, approximately 400 mg of each formula was mixed into deionized water to make 1.5 mL suspensions in 1.5-mL polypropylene microcentrifuge tubes with snap caps. Following sonication in a warm water bath (30–40 °C) for 15 min, the suspensions were allowed to settle for 5 min. From the light-coloured top portion of the suspension, a few drops of liquid were transferred to the wells of an aluminum cryo-SEM holder, where a small amount was allowed to overfill. The cryo-SEM holder with sample was then submerged in slushy liquid nitrogen for 10–20 s to rapidly freeze. After freezing, the sample was vacuum transferred into a Quorum PP3010T cryochamber (Quorum Technologies, East Sussex, UK) to fracture the overfill portion and reveal a cross section of the frozen sample. The fractured sample was then transferred to a Helios NanoLab 650 scanning electron microscope (FEI Company, Hillsboro, OR, USA) for imaging. Cryo-SEM images were collected with a current of 13 pA at 2 kV, with a working distance of 4 mm, at a scanning resolution of 3072 × 2207 or lower by averaging 128 low dose scanning frames with drift correction. The sample was kept at −140 °C when fracturing and imaging. The sample was also imaged after sublimation at −80 °C for 15 min in the cryo-SEM chamber to remove residual water. In the cryo-SEM image, the approximate diameters of particles-of-interest were measured using ImageJ software as described in the software instructions, similar to length measurements performed by Lam et al. [40,41].

*2.7. Zeta Potential*

The most promising formula (LM3) was prepared for zeta potential measurements. A total of 400 mg of the formula was combined with 10 mL of ultrapure water (DI water, 18.2 MΩ-cm) in a 15 mL centrifuge tube, vortexed and sonicated, and then centrifuged at 3800 rpm. From the resulting supernatant, 200 μL was transferred into a separate 15 mL centrifuge tube and diluted with ultrapure water to a total volume of 10 mL. The ultrapure water was obtained using a Barnstead Nanopure water purification system (Thermo Fisher Scientific Inc., MA, USA), We transferred the diluted solution into a clean Malvern DTS1060 folded capillary cell (Malvern Panalytical Inc., MA, USA) for zeta potential measurement. A Malvern Zetasizer Nano ZS instrument (Malvern Panalytical Inc., MA, USA) was used to collect measurements at 25 °C within 15 min of sample preparation. The measurements were performed in triplicate using automatic attenuation selection and automatic voltage selection. The pH of the diluted solution was also monitored using a IQ240 Portable pH Meter (IQ Scientific Instruments, Carlsbad, CA, USA).

*2.8. Data Analysis*

Results were reported as means ± standard errors of the mean (SEM). Statistical comparison was performed with two-way ANOVA followed by post hoc Bonferroni correction for solubility measurements and Kruskal–Wallis test with post hoc Dunn's correction for permeability experiments. Differences between sample groups were considered statistically significant at $p < 0.05$.

## 3. Results

*3.1. Solubility Measurements*

All novel formulations exhibited higher solubilities when compared to baseline control Vitamin $D_3$ (BC, Table 2, Figure 2). LM1 proved to be more soluble than the other formulations in all tested media, and showed approx. 5 times better solubility than the control when tested under intestinal conditions ($p < 0.05$; Table 2). Figure 2 provides a graphical comparison of the solubilities where differences in solubility were apparent.

**Table 2.** Solubilities of Vitamin $D_3$ formulations in different aqueous media measured with UHPLC.

| | Formula LM1 | Formula LM2 | Formula LM3 | Formula BC |
|---|---|---|---|---|
| Water (pH 6.3) | $7.55 \times 10^{-4} \pm 2.81 \times 10^{-4}$ | $2.89 \times 10^{-4} \pm 1.27 \times 10^{-5}$ | $1.97 \times 10^{-4} \pm 4.20 \times 10^{-5}$ | $1.64 \times 10^{-4} \pm 2.57 \times 10^{-5}$ |
| Gastric juice (pH 1.2) | $6.93 \times 10^{-4} \pm 1.31 \times 10^{-4}$ | $4.22 \times 10^{-4} \pm 4.34 \times 10^{-5}$ | $3.37 \times 10^{-4} \pm 1.76 \times 10^{-4}$ | $2.40 \times 10^{-4} \pm 4.86 \times 10^{-5}$ |
| Intestinal juice (pH 6.8) | $1.09 \times 10^{-3} \pm 5.09 \times 10^{-4}$ [a] | $7.19 \times 10^{-4} \pm 1.90 \times 10^{-4}$ [ab] | $2.89 \times 10^{-4} \pm 7.92 \times 10^{-4}$ [b] | $1.91 \times 10^{-4} \pm 4.24 \times 10^{-5}$ [b] |

Values displayed as milligrams of cholecalciferol per millilitre of water (mg/mL); n = 3; $p < 0.05$ (two-way ANOVA, post hoc Bonferroni correction between "a" and "b").

*3.2. Permeability Measurements*

The $P_{app}$ values from Caco-2 monolayer tests suggested a significantly higher permeability for LM3 ($1.6 \pm 0.3 \times 10^{-5}$ cm/s; $p < 0.01$; Table 3) in comparison with LM1, LM2 and BC ($1.9 \pm 0.3 \times 10^{-7}$ cm/s, $3.6 \pm 0.8 \times 10^{-6}$ cm/s and $1.3 \pm 1.1 \times 10^{-9}$ cm/s, respectively; Table 3). Figure 3 shows the UHPLC chromatograms for the determination of Vitamin $D_3$, and Figure 4 shows a graphical representation of the permeabilities of these formulations. Furthermore, different investigators adopted slightly different criteria for ranking a compound as having either "high permeability" or "low permeability". For example, some viewed a $P_{app}$ value that is more than $1 \times 10^{-7}$ cm/s as highly permeable [42], while Wahlang and colleagues cited $1.4 \times 10^{-7}$ cm/s as a highly permeable $P_{app}$ value and $5 \times 10^{-6}$ cm/s as a lowly permeable $P_{app}$ value [43]. Fossati et al. considered $3 \times 10^{-6}$ cm/s as the delimiting value, where a $P_{app}$ value less than $3 \times 10^{-6}$ cm/s would belong in the low permeability group and a $P_{app}$ greater than that value would be ranked as highly permeable [44]. Despite the somewhat diverse criteria, formula LM3 demonstrated a high degree of permeability with its $P_{app}$ value on the order of $10^{-5}$ cm/s. However,

depending on the criteria used, LM2 could be considered fairly or slightly permeable, while LM1 and BC would be considered as having low permeabilities since their $P_{app}$ values were orders of magnitude lower than the $10^{-6}$ cm/s range.

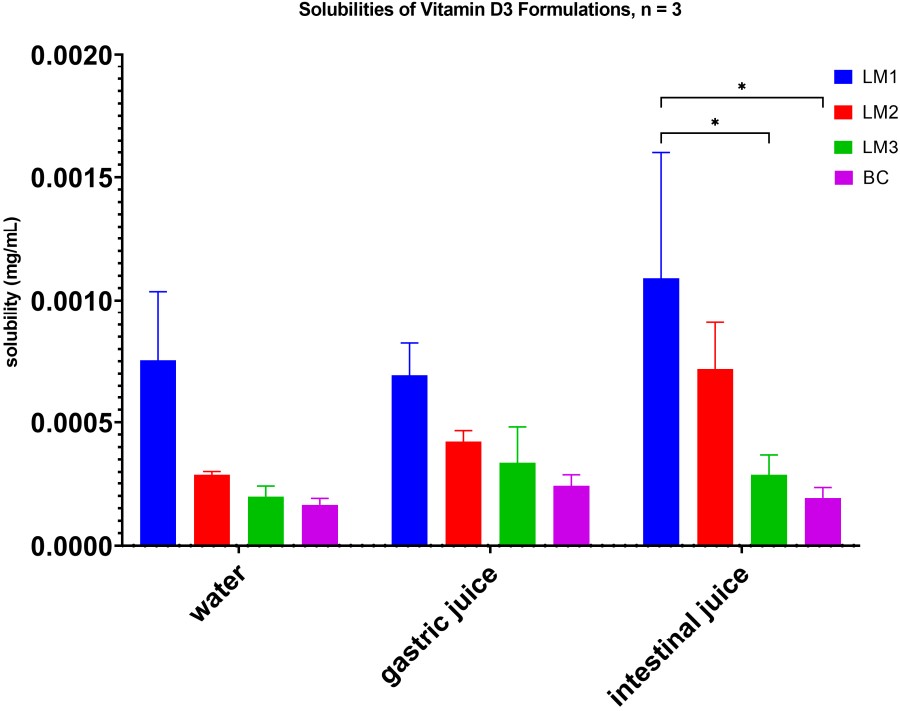

**Figure 2.** Solubility measurements of the four Vitamin $D_3$ formulations. Taller bars correspond to higher levels of dissolved Vitamin $D_3$, suggesting increased solubility. LM1 demonstrated significantly higher solubility under all 3 conditions tested compared to the control (BC); * $p < 0.05$ (two-way ANOVA, post hoc Bonferroni correction).

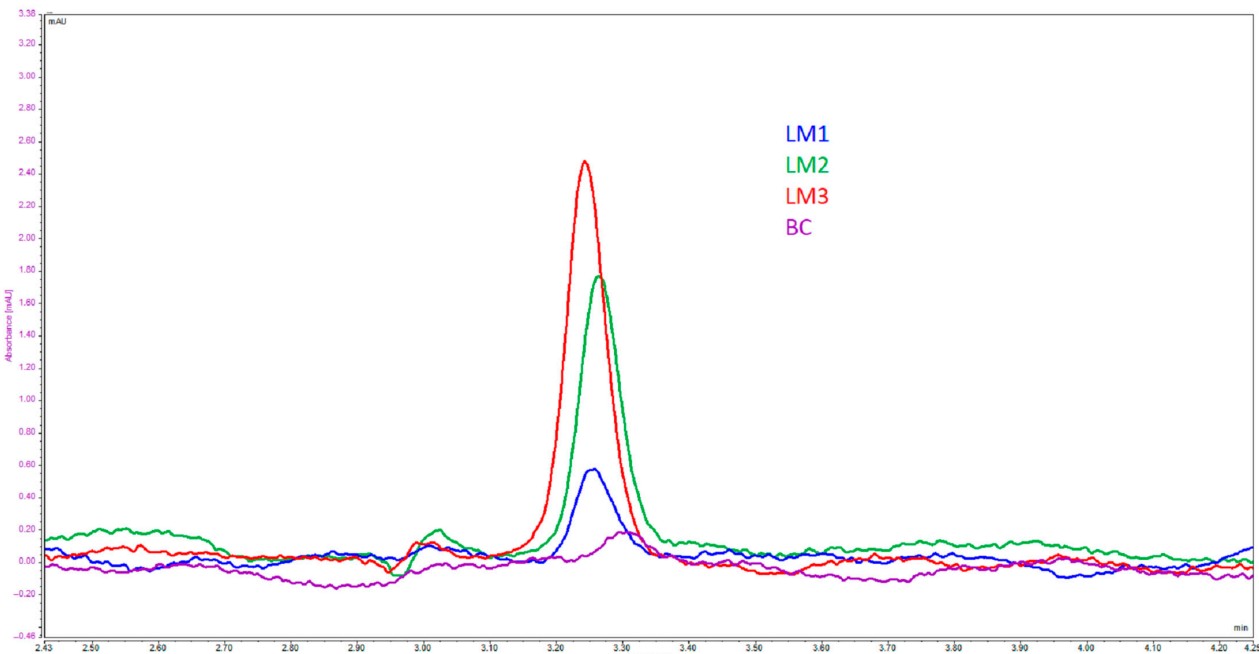

**Figure 3.** UHPLC chromatograms of the basal solutions collected from permeability experiments. The primary peak in the chromatogram represents Vitamin $D_3$ and concentrations were calculated using measured peak areas against an external calibration standard.

**Table 3.** Permeabilities of the four Vit D formulations.

| Formula LM1 | Formula LM2 | Formula LM3 | Formula BC |
|---|---|---|---|
| $1.9 \pm 0.3 \times 10^{-7}$ [ab] | $3.6 \pm 0.8 \times 10^{-6}$ [ab] | $1.6 \pm 0.3 \times 10^{-5}$ cm/s [b] | $1.3 \pm 1.1 \times 10^{-9}$ [a] |

n = 4 per treatment. [a,b] Means in a row without a common superscript letter differ by $p < 0.05$, as analyzed by Kruskal–Wallis test followed by Dunn's correction for multiple comparisons.

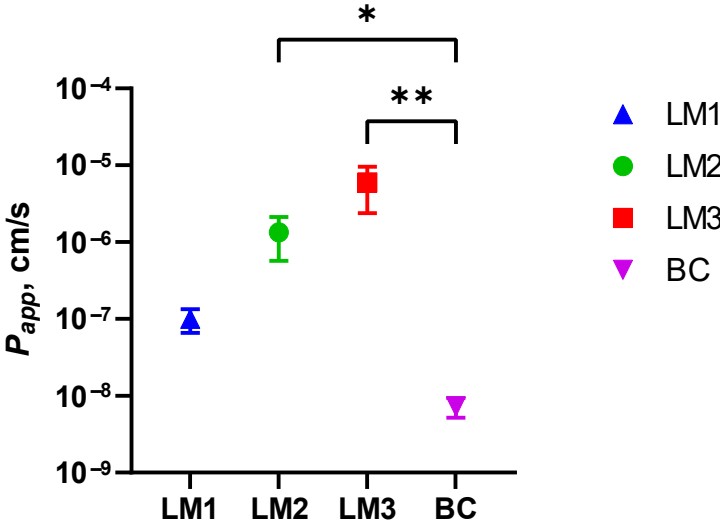

**Figure 4.** Logarithmic plot of the permeabilities of the four Vitamin $D_3$ formulations. The X-axis lists each formula code, while the Y-axis represents $P_{app}$ values measured in cm/s. n = 4 per formula. Higher data points along the Y-axis represents increased permeability. LM3 achieved significantly higher permeability compared to the baseline control; * $p < 0.05$, ** $p < 0.01$ (Kruskal–Wallis test with Dunn's correction for multiple comparisons).

### 3.3. Laser Diffraction, Micellar Morphology, Size and Stability

Laser diffraction measurements are taken in triplicates for each formulation, and the average of the three measurements are presented in Figure 5 and Table 4. Only LM3 has been further analyzed regarding its morphology and stability features due to the promising Caco-2-cell permeability results (Table 3).

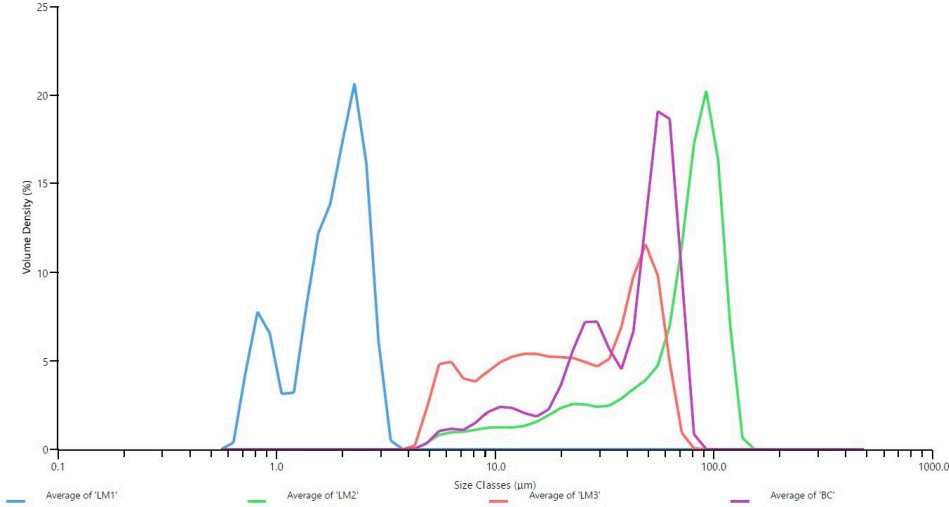

**Figure 5.** Particle size distribution of the formulations as determined by Laser diffraction. Each formulation was measured in triplicates and the average volume density was graphed against the average particle size. Volume density is a measure of the proportion of particles present at each size interval based on their volume, and the size classes represent the particle size in micrometres.

**Table 4.** Summary of particle size distribution and PDI of the four formulations. $D_{N\%}$ reflects the percentage of particles (N%) with a diameter (D) less than or equal to the specified value [46].

|  | Formula LM1 | Formula LM2 | Formula LM3 | Formula BC |
|---|---|---|---|---|
| $D_{10\%}$ (μm) | 0.864 | 17.7 | 6.65 | 12.7 |
| $D_{50\%}$ (μm) | 1.89 | 77.1 | 23.4 | 46.4 |
| $D_{90\%}$ (μm) | 2.62 | 107 | 54.2 | 66 |
| PDI | 0.617 | 1.59 | 0.315 | 1.13 |

As for LM1, the smaller particle size likely corresponds to the higher solubility in the tested media (Table 2) when compared to the other formulations. For LM3, the particle sizes ranged from a few micrometres to larger than 50 micrometres, with a PDI slightly larger than 0.3 (Table 4). This agrees with the cryo-SEM evidence, where we observed spherical particles ranging from approximately 1 or 2 μm to 57 μm in diameter (Figure 6). This size range is larger than nanoparticular preparations of Vitamin D, which were typically observed in the nanometre range and indicates that LM3 is a polydisperse formula [25,34,38,39,45].

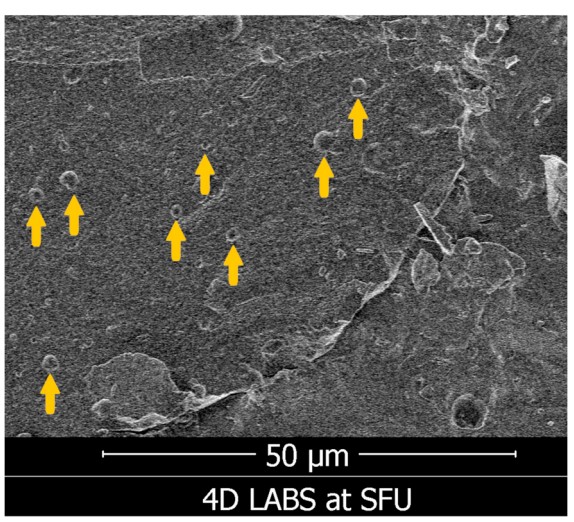

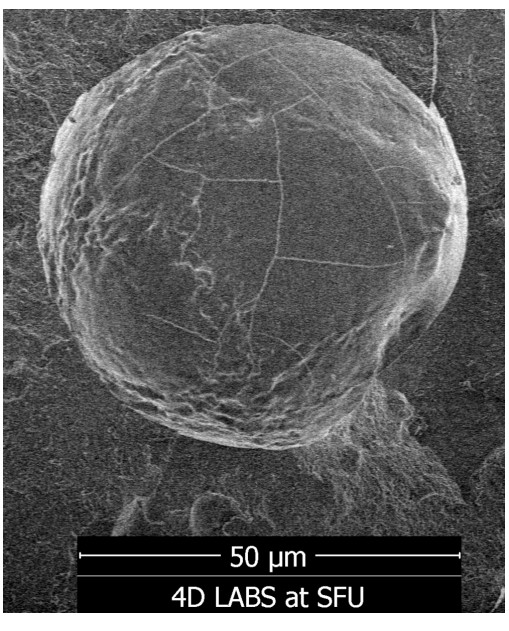

(**a**) (**b**)

**Figure 6.** Appearances of LM3 particles captured with cryo-SEM. The particles were generally spherical in shape, with diameters in the micrometre range, as measured with the ImageJ software using the "50 μm" scale bar for reference. (**a**) Yellow arrows indicate examples of LM3 particles, which were approximately 1 to 2 μm in diameter. (**b**) An example of a larger LM3 particle approximately 57 μm in diameter.

Additionally, the average zeta potential of LM3 was $-51.56 \pm 2.76$ mV at pH 5.83, which was indicative of colloidal stability under these conditions in aqueous solution (Figure 7).

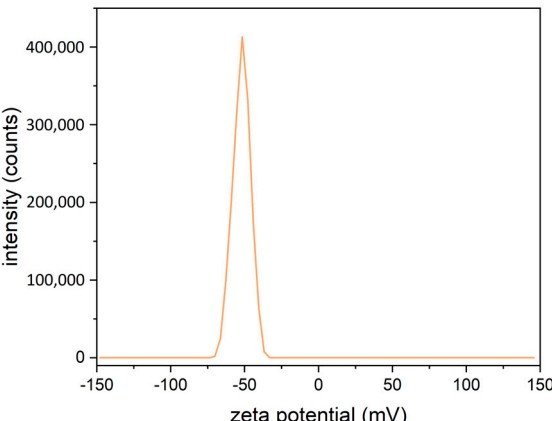

**Figure 7.** Graphical representation of the average zeta potential of LM3, measured in triplicates at a pH of 5.83. The average zeta potential was $-51.56$ mV $\pm$ 2.76 mV. The error in this study is reported as three times the standard deviation of the calculated mean.

## 4. Discussion

In this study, we examined the properties of three novel LipoMicel® formulations (LM1, LM2 and LM3), hypothesizing that at least one formula would demonstrate advantageous properties for intestinal absorption. The best-performing candidate in terms of intestinal permeability could be selected for subsequent in vivo experiments. We evaluated their potential effectiveness to deliver Vitamin $D_3$, and observed that LM3 was the most promising formula. Hereafter, "LipoMicel" (or "LipoMicel Vitamin $D_3$") will specifically refer to LM3.

Significantly, Caco-2 permeability experiments showed that LipoMicel® is 12,000 times more permeable compared to the baseline control (Table 3 and Figure 4). This is several orders of magnitude higher than the 2.5 times enhanced uptake reported on lysophosphatidylcholine micelles [30], the 4-fold increase in bioavailability reported in a casein micelles [47] and the 5.3-fold increase reported in pea-protein-based nanoemulsions [48]. However, in this study it appears that the increased permeability of micellar Vitamin $D_3$ does not correlate with increased solubility or reduced particle size—since LipoMicel has similar solubility characteristics (Figure 2) and particle size distribution (Figure 5 and Table 4) as the baseline control. While other reports on Vitamin D micelles focused on structural morphology, stability, bioaccessibility and controlled release without reporting permeability [29,49,50], this study demonstrates that biological experiments provide better insights on Vitamin D absorption than simple diffusion modelled by physical and chemical characterization of the formulations [12]. In fact, the results of our study suggest that the absorption process involves biological mechanisms such as cholesterol transporters and cell-membrane efflux pumps such as ABCB1, which would support previous findings [32,51]. However, this assumption needs to be further evaluated in future studies.

LipoMicel was further characterized to provide data for reproducing similar biological effects in future formulations. Cryo-SEM of LipoMicel showed spherical particles in the low micrometre size range (for example, numerous particles were approximately 1 to 2 μm in diameter), as well as larger ones exceeding 50 μm in diameter. In general, liposomal drug delivery vehicles can range from 50 to 200 nm in diameter [52,53] and micellar ones can approach 1 μm [54]. Micrometre-sized carriers have been proven to be physically advantageous for cellular uptake. For example, using a rat as an animal model, a study found that spherical particles up to 5 μm in diameter can be absorbed through the intestines when locally delivered, and those up to 1 μm can be absorbed when orally administered [55]. Other delivery systems (such as microemulsions or some microcapsules) can even exist at sizes up to 5000 μm in diameter [35,56]. Such systems can be referred to as microcapsules, defined by Vieira and Souza as having diameters between 1–5000 μm [56]. The current iteration of LipoMicel falls under this category.

The thermodynamic stability of such beyond-nanoscale structures can allow them to be more easily manufactured than nanopreparations such as liposomes [35,56]. Furthermore, compared to nanoscale liposomal formulations, microcapsules may be more resistant to environmental stresses such as changes in temperature and pH [35,56]. Additionally, microcapsules have been observed to achieve better bioavailability of Vitamin D by way of improving the vitamin's controlled and targeted release [56]; this is not in disagreement with the drastic increase in permeability when LipoMicel was introduced to Caco-2 cells. As such, the ease of manufacture, thermodynamic stability and potentially improved bioavailability are definitive advantages of microscale formulations, especially considering the need to mass-produce health supplements for the consumer market. We aim to conduct follow-up pharmacokinetic studies to determine how LipoMicel, as a microencapsulated formulation using natural and safe ingredients, compares to other delivery systems such as liposomes and nanoemulsions in the context of enhanced uptake in vivo.

With a PDI value slightly larger than 0.3 (Table 4), LipoMicel® appears to be slightly polydisperse in solution, albeit nearly monodisperse according to the definition mentioned by Danaei et al. [38]. When viewed in conjunction with the particle size distribution pattern (Figure 5), we see that although there's considerable size variation, the majority of the LipoMicel particles may be approximately 50 μm in diameter. Worth noting is that the complete size range of LipoMicel is almost identical to that of the baseline control, but the two formulations demonstrate drastically different permeability results. Smaller LipoMicel particles do exist, as revealed by cryo-SEM. The aggregation of such particles could contribute to the natural formation of their larger counterparts. However, the larger ones may not be the primary carriers of Vitamin $D_3$ for absorption since the bulky topology would present challenges in crossing the mucus lining, be inefficient at contacting the intestinal epithelium and be easily subjected to the actions of bile salts and digestive lipase. Under physiological conditions, the constant peristaltic agitation in combination with enzymatic efforts could break some larger particles into smaller ones, which would then become the primary carriers and the agents of enhanced absorption. To confirm whether the constant digestive actions would increase the number of uniformly small-sized LipoMicel particles, future experiments are warranted. Furthermore, since different excipients at varying ratios could alter the PDI of a formulation, future efforts may be directed at adjusting the composition of LipoMicel so that it could reach lower PDI, become increasingly monodisperse at a smaller particle size (within the micrometer range) and achieve homogeneity [46,57]. A monodispersed, smaller-sized version of LipoMicel could improve bioavailability even further and stands to be explored in subsequent studies.

At an average zeta potential of −51.56 mV, LipoMicel was stable in aqueous solutions, where values with magnitudes greater than 30 mV are considered stable [58]. Referencing the criteria of some investigators, LipoMicel could be further categorized as belonging to a group with "good stability" [59]. The measured zeta potential for LipoMicel is comparable with some novel liposomal preparations of Vitamin $D_3$ (for example, those with zeta potentials in the −50 mV to −70 mV range), while considerably exceeding other preparations (for example, at −26.70 mV or −4.0 mV) [60–62]. Moreover, although zeta potential is not a direct assessment of surface charges, having a negative zeta potential could imply that the LipoMicel also has a negative surface charge [63]. Certainly, pH and other factors can affect zeta potential measurements, so the exact surface charge stands to be determined in future experiments [58]. However, if this is true, then this property could facilitate absorption since some have suggested that having a negative surface charge prevents the particles from adhering to intestinal mucus, which is also negatively charged [64]. These particles could then penetrate the intestinal mucosal layer and reach the absorption surface at a quicker pace.

Despite the mechanistic complexity of drug absorption in humans, we can gain a measure of understanding on a formula's bioavailability potential when its solubility and permeability in vitro are known [65]. In the case of Vitamin $D_3$, its natural hydrophobicity renders it insoluble in the aqueous intestinal environment and this can create challenges

for its permeability across enterocytes. Theoretically, packaging a hydrophobic compound in amphipathic micellar structures can potentially overcome such challenges. Some have hypothesized that taking Vitamin $D_3$ with high-fat meals can facilitate the production of natural micelles through the assistance of bile [66]. However, the types and amounts of dietary fat vary in their effects on Vitamin $D_3$ absorption [66]. Furthermore, bile production between individuals could vary. In this regard, it is advantageous to have measured control over micelle formation to achieve both high solubility and high permeability through product formulation instead of relying on the behaviour and physiology of end users.

The current LipoMicel formula showed similar but slightly improved solubility characteristics (i.e., in water and gastrointestinal media) compared to the control. Although LipoMicel was less soluble in comparison with the other formulations (LM1 and LM2), its $P_{app}$ value can successfully qualify the current formula for consideration as a highly permeable drug in vitro, suggesting that once the particles reach enterocytes, they can readily cross the gut–blood barrier to ferry the cargo (Vitamin $D_3$) for subsequent transport.

Indeed, we know that drugs with low or moderate solubility but high permeability can achieve complete absorption provided that they overcome the dissolution challenge [67]. As such, to decidedly test how the current LipoMicel® formula would perform in the human body, subsequent in vivo studies with human subjects are still necessary. In addition, formula manipulation remains a useful tool for further enhancing its solubility and permeability [68]. Such efforts could aim at altering the existing excipients, or examining the possibilities of creating nanoemulsions, self-emulsifying drug delivery systems, solid dispersion, or other methods of delivery [28].

During our study, we used Caco-2 as an in vitro cell model. While Caco-2 is a common model used to study the absorption and transport of nutrients in the small intestine, studies on Vitamin $D_3$ transport across Caco-2 cell monolayers are rather limited. Our work reaffirms that this robust cell model is an effective tool for studying Vitamin $D_3$ permeability. Although others have investigated various micellar preparations of Vitamin D using Caco-2 cells, the formulations they used may not be practical in the context of over-the-counter health supplements [12]. For example, depending on the regulatory jurisdiction and context, certain ingredients used in those preparations would not be permitted in health supplement products. Such ingredients include lysophosphatidylcholine, monoolein, taurocholate and fetal bovine serum [12]. At the time of this writing, they are not allowed in an oral health supplement marketed in Canada. In contrast, our work examined formulations made from natural, safe ingredients which could be practically marketed.

Certain limitations do exist when using Caco-2 cells for nutrient absorption studies. For instance, the cells are not a perfect model of the human small intestine. Caco-2 cells form a monolayer of a single cell type. While they are adept at expressing a range of membrane transporters and certain junctions similar to those found in the small intestine, their limited differentiation would not fully represent all epithelial traits relevant to absorption [69].

The in vitro data obtained from this study characterized a micellar form of Vitamin $D_3$, as LipoMicel Vitamin $D_3$. This novel formula displays advantageous morphological and physicochemical traits and provides a significant improvement in intestinal permeability when measured in vitro. Although these results are promising, we recognize that it is never a straightforward extrapolation from in vitro characteristics to in vivo bioavailability. Therefore, more work is needed to determine if the current iteration of LipoMicel can achieve improved bioavailability in vivo. To this effect, current efforts are underway to examine the bioavailability of LipoMicel Vitamin $D_3$ in human volunteers.

## 5. Patents

Pending for LipoMicel® Matrix—Eutectic Matrix for Nutraceutical Compositions lists inventors as: R.J.G., S.W., Y.C.K. and C.C. No inventor benefits from this and the ownership belongs to InovoBiologic Inc (Calgary, AB, Canada).

**Author Contributions:** Conceptualization, C.C., J.S. and M.D.; methodology, X.Z., Y.Z., M.J.R. and Y.C.K.; formal analysis, Y.Z. and Y.C.K.; investigation, X.Z., Y.Z., M.J.R. and Y.C.K.; resources, C.C., R.J.G., X.Z. and M.J.R.; data curation, X.Z., Y.Z., M.J.R., Y.C.K. and C.C.; writing—original draft preparation, M.D.; writing—review J.S., S.W. and M.D.; visualization, X.Z., Y.Z. and Y.C.K.; supervision, C.C. and J.S.; project administration, C.C. and J.S. All authors have read and agreed to the published version of the manuscript.

**Funding:** Open access funding was provided by ISURA's research fund as part of its non-profit mandate.

**Institutional Review Board Statement:** Not applicable.

**Informed Consent Statement:** Not applicable.

**Data Availability Statement:** Data are contained within the article.

**Acknowledgments:** This work made use of the 4D LABS shared facilities supported by the Canada Foundation for Innovation (CFI), British Columbia Knowledge Development Fund (BCKDF), Western Economic Diversification Canada (WD), and Simon Fraser University (SFU). We thank Byron Gates and the Gates Research Group for providing the access, training and resources for collection and analysis of the zeta potential data. We thank Acacia Chang for her work on Figures 1 and 4 as well as the graphical abstract. LipoMicel® is the registered trademark of Natural Factors Group. The Factors Group of Nutritional Companies supplied the test samples.

**Conflicts of Interest:** The authors M.D., C.C., Y.Z., Y.C.K. and J.S. are employees of Isura. Isura is a not-for-profit organization. R.J.G. is the owner of the Factors Group of Companies, S.W. is a consultant to InovoBiologic Inc. (Calgary, AB, Canada). The authors declare no other conflict of interest.

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
