# Peer review of "Designing Vitamin D3 Formulations: An In Vitro Investigation Using a Novel Micellar Delivery System"

_nutraceuticals, doi:10.3390/nutraceuticals3020023_

Round 1

Reviewer 1 Report

Comments and Suggestions for Authors

I critically evaluated the article entitled " Designing Vitamin D3 formulae: an in vitro investigation using a novel micellar delivery system ". The topic is interesting. However, there are several critical points in the manuscript before publication.

What is the novelty of the formulation? According to the title, you should design a novel formulation for the delivery of vitamin D3.

The number of keywords is more than standard.

Your results do not have any statistical analysis.

The sample preparation was not clear (section 2.1).

Your measurements on the liposome sample are not enough. You should investigate particle size by the DLS method, storage stability in terms of changes in particle size and zeta, and encapsulation efficiency of Vitamin D.

No variable has been considered for the experimental design and only the characteristics of one sample have been examined. 

Author Response

We would like to thank the reviewer for the constructive feedback and taking the time to review this manuscript. Please find below – our responses to address these comments.

Reviewer 2 Report

Comments and Suggestions for Authors

Manuscript  is well written. However, some points need to be resolved

1. Solubility of Vitamin D3 has been found very low in all formulations. So, This need to be justified properly. If equilibrium solubility studies has been conducted in all ingredients data is must in manuscript. 

2. Grammar needs to improve.

3. Stability data of final formulation needs to be included.

4. UHPLC peak and stability indicating methods with results can be included.

5. Introduction can be well written. why micellar formulation is required is not coming out properly and previous studies also need to be reported. 

Comments on the Quality of English Language

Grammar needs to improve in abstract as well as in paper. 

Author Response

(The authors gave the same response as above.)

Reviewer 3 Report

Comments and Suggestions for Authors

The manuscript titled "Designing Vitamin D3 formulae: an in vitro investigation using a novel micellar delivery system" studies a micellar form to incorporate Vitamin D3 into LipoMicel, which shows the potential to promote stability and permeability of Vitamin D3. However, it did not acheive enhanced solubility as expected, and need futher explanations on data conclusion and experimetal design. Therefore, it requires major revisions for the purpose of publication.

Question 1: In Table 1, it is confusing that superscript 1 corresponds to two compounds: Medium chain triglycerides and Flaxseed oil. Please update. If the flaxseed oil is the carrier oil, does it suppose to be in other formulas?

Question 2: In Table 1, 3 formulae were prepared by adding different excipients. What are the molar ratios of each excipient? Will the molar ratio from an excipient impact the formulation or Vitamin D3 encapsulation? Please comment.

Question 3: Line 255 Isn't LM3 have a high permeability according to the criteria? Please double-check to confirm the data agrees with the statement.

Comment 1: After the sentence of lines 80-82, please add 1-2 sentences to describe the LipoMicel matrix, and how it generally works to improve absorption and efficacy.

Comment 2: It is known that micellar formulation can encapsulate hydrophobic compounds. The low solubility may be from the preparation procedure. A 0.45 μm PTFE filter is used but the overall size of the micelle is 1-2 μm from Figure 4 SEM image, so it is highly possible that Vitamin D3 in LipoMicel was lost during the preparation. Two experimental designs are recommended:

1. prepare BC and formulae without filtering, and compare the solubility of Vitamin D3 again.

2. perform DLS measurements to understand the overall size and distribution of micelles in aqueous solution.

Comment 3: Add the CMC (critical micelle concentration) for Lipomicel.

Author Response

(The authors gave the same response as above.)

Round 2

Reviewer 1 Report

Comments and Suggestions for Authors

·       The specifications of all used materials should be added before sample preparations.

·       As I previously mentioned, the preparation of different formulations is unclear. The concentration of material? mixing conditions in terms of temperature, speed, the equipment? Please explain the process clearly.

·       The number of methods should be edited after the addition of particle size measurement.

·       3.3. The particle size results are shown in Fig. 5 but with no discussion. Moreover, the explanation of zeta measurement is mistakenly deleted in the revised manuscript. Please compare the result of particle size reported by DLS with cryo-SEM and explain the reasons for this remarkable contradiction results.

·       According to particle size measurement, all formulations are not in the nanoscale range. Please explain the advantages of these formulations (in terms of bioavailability, permeability, efficiency, and stability ) compared to other nano-delivery systems like nanoliposome, nanoemulsion, SLN, NLC, etc.  

Reviewer 2 Report

Comments and Suggestions for Authors

I am concerned as many studies are reported on reported on drug delivery of Vitamin D using micellar formulation. Thus, topic is not innovative. 

In particle size distribution study particles of micellar formulation appears to be polydisperse. and PDI also not included in manuscript anywhere.

Solubility studies results have been presented in complicated way. 

Comments on the Quality of English Language

Revision required

Reviewer 3 Report

Comments and Suggestions for Authors

The quality of the manuscript was improved after edits. However, it is expected to add what is the number of the molar ratio of excipients in the formulations to inform the audience: Table 1 regarding my Question 2.

Also, the author's reply of findings on "the impact was relatively minor in comparison to the impact from addition of different excipients." can also be added in the text to adequately discuss. 
